# From Kidney Protection to Stroke Prevention: The Potential Role of Sodium Glucose Cotransporter-2 Inhibitors

**DOI:** 10.3390/ijms24010351

**Published:** 2022-12-26

**Authors:** Cheng-Yang Hsieh, Sheng-Feng Sung

**Affiliations:** 1Department of Neurology, Tainan Sin Lau Hospital, Tainan 701, Taiwan; 2School of Pharmacy, Institute of Clinical Pharmacy and Pharmaceutical Sciences, College of Medicine, National Cheng Kung University, Tainan 701, Taiwan; 3Division of Neurology, Department of Internal Medicine, Ditmanson Medical Foundation, Chia-Yi Christian Hospital, Chiayi City 600, Taiwan; 4Department of Beauty & Health Care, Min-Hwei Junior College of Health Care Management, Tainan 736, Taiwan

**Keywords:** stroke, chronic kidney disease, diabetes, sodium glucose cotransporter 2 inhibitors, SGLT2 inhibitors

## Abstract

Chronic kidney disease (CKD) is an independent risk factor for stroke and covert cerebrovascular disease, and up to 40% of stroke patients have concomitant CKD. However, the so-called “cerebrorenal interaction” attracted less attention compared to its cardiorenal counterpart. Diabetes is the leading cause of CKD. The sodium–glucose cotransporter (SGLT) 2 inhibitor is a relatively new class of oral anti-diabetic drugs and has cardiorenal benefits in addition to glucose-lowering effects. In the present perspective, we would like to review the current status and future potential of the SGLT2 inhibitor in cerebro–renal interactions and strokes regardless of the status of diabetes. We propose the potential roles of baseline renal functions and SGLT1/2 dual inhibition in stroke prevention, as well as the additional benefits of reducing atrial fibrillation and hemorrhagic stroke for SGLT2 inhibitors. Further clinical trials are anticipated to test whether SGLT2 inhibitors can fulfill the long-standing unmet clinical need and stop such a vicious cycle of cerebro–renal interaction.

## 1. Introduction

The global prevalence of chronic kidney disease (CKD) increased by approximately one-third since 1990, reaching 9.1% in 2017 [1]. Furthermore, mounting evidence suggests that CKD may be an independent risk factor for stroke [2,3,4,5,6,7,8] and covert cerebrovascular diseases [9,10]. On the other hand, up to 40% of stroke patients have a concomitant CKD [11]. Given the increasing burden of CKD in the global ageing society and the increased risk of stroke in CKD, it is anticipated that physicians will encounter more and more stroke patients with CKD. However, in the past, there was a very limited number of pharmaceutical agents available for use in this clinical scenario [3]. 

Diabetes is the leading cause of CKD and diabetic kidney disease accounts for about one-third of disability-adjusted life-years lost due to CKD [1]. As a relatively new class of oral anti-diabetic drugs, sodium–glucose cotransporter-2 (SGLT2) inhibitors are attracting more attention nowadays due to their pleiotropic effects [12,13]. Via the inhibition of SGLT2, SGLT2 inhibitors decrease blood glucose levels by reducing glucose reabsorption in proximal renal tubules, and they promote natriuresis and osmotic diuresis. In addition to the glucose-lowering effect, they have been shown to preserve renal function [14] and reduce hospitalization for heart failure [15] irrespective of the diabetes status [16]. The role of SGLT2 inhibitors in the treatment of cardiorenal syndrome has now been well established [12,13,17,18,19].

However, whether SGLT2 inhibitors play a role in preventing stroke and cerebrovascular disease is still controversial. Meta-analyses show that SGLT2 inhibitors as a class have a neutral effect on the risk of strokes in diabetic patients [20,21]. On the contrary, real-world observational studies demonstrated the association between the use of SGLT2 inhibitors and a decreased risk of stroke [22]. Notably, previous dedicated cardiovascular outcome trials (CVOTs) for SGLT2 inhibitors have mainly enrolled diabetic patients with established cardiovascular diseases other than stroke [23]. In addition, a non-fatal stroke is typically considered a component of the composite outcome of major adverse cardiovascular events (MACE) rather than the primary endpoint [24]. Therefore, these CVOTs may have insufficient statistical power for testing the effect of SGLT2 inhibitors on stroke risk reductions. Moreover, stroke physicians and neurologists may not be familiar with SGLT2 inhibitors, which have the potential of being incorporated into their stroke risk reduction armamentarium.

Therefore, in the present perspective, we would like to review the current status and future potential of SGLT2 inhibitors in cerebro–renal interactions and strokes.

## 2. SGLT2 Inhibition and Kidney Protection

Via the inhibition of SGLT2 in the proximal tubules, the concentration of sodium in the glomerular filtrate increases, thereby activating the tubuloglomerular feedback and causing the vasoconstriction of afferent arterioles. In addition, the macula densa also inhibits the release of renin from the juxtaglomerular cells and enhances the dilatation of efferent arterioles. As a result, the intraglomerular hydrostatic pressure reduces, bringing on the renoprotective effect of SGLT2 inhibitors [13]. 

Recently, a meta-analysis of 13 large, randomized trials showed that SGLT2 inhibitors reduced the risk of kidney disease progression by 37% [14]. The effect was universal irrespective of diabetes status or primary kidney disease. Since CKD can precipitate the development of cerebrovascular disease, it is worth examining whether strokes can be prevented via kidney protection by SGLT2 inhibition.

## 3. Cerebro–Renal Interaction

The definition of CKD comprises an estimated glomerular filtration rate (eGFR) that is less than 60 mL/min/1.73 m^2^ and/or evidence of renal injury (e.g., albuminuria) that persists for at least three months [25]. Meta-analyses have shown that either a decreased GFR or the presence of albuminuria is independently associated with incident stroke [26,27]. In addition, CKD is associated with covert cerebrovascular diseases [9]. There are multiple interrelated mechanisms underlying these associations. First, several traditional risk factors, such as hypertension and diabetes, are shared between CKD and stroke. Second, CKD-specific risk factors may further increase the risk of strokes, such as the accumulation of uremic toxins as well as vascular medial calcification and endothelial damage due to abnormal phosphorus–calcium metabolism [6,7,28]. Third, CKD may precipitate certain stroke risk factors such as atrial fibrillation [29]. On the other hand, stroke or other brain injuries may subsequently cause renal dysfunctions [30]. However, the so-called “reno-cerebrovascular disease” [2], or cerebro–renal interaction [4,30], attracted less attention compared to its cardiorenal counterpart [18].

Patients with CKD are prone toward thrombotic events. However, they also tend to experience major bleeding such as intracerebral hemorrhage (ICH) [3,4,6] when treated with antiplatelet agents, anticoagulants, or thrombolytic agents. The Gordian knot of an increased risk of both thrombosis and bleeding long puzzled clinicians who tried to untie it. Furthermore, patients with advanced CKD are typically excluded from clinical trials of new cardiovascular drugs [28]. Consequently, evidence-based treatments are often lacking for patients with CKD, a population that is highly vulnerable toward cerebrovascular and cardiovascular morbidity and mortality.

## 4. Stroke Risk with SGLT2 Inhibitors: Evidence from Existing CVOTs

In 2008, the US Food and Drug Administration mandated that all new anti-diabetic drugs should undergo an assessment of cardiovascular safety before obtaining approval [31]. Important patient demographics, variables, and cerebrovascular outcomes in the previous CVOTs of SGLT2 inhibitors are summarized in Table 1.

In the EMPA-REG OUTCOME trial (Empagliflozin Cardiovascular Outcome Event Trial in Type 2 Diabetes Mellitus Patients—Removing Excess Glucose), Empagliflozin has been shown to cause a numerically higher risk of both non-fatal stroke (hazard ratio (HR), 1.24; 95% confidence interval (CI), 0,92–1.67) and total (fatal or non-fatal) stroke (HR, 1.18; 95% CI, 0.89–1.56) compared to the placebo, although the results did not reach statistical significance [32]. Possible mechanisms for such observations might include postural hypotension or hemoconcentration [33,34] or the result of chance alone. The subsequent three CVOTs showed that canagliflozin, dapagliflozin, and ertugliflozin all had a neutral effect on strokes [35,36,37]. However, canagliflozin seemed to be associated with a lower risk of hemorrhagic stroke (HR, 0.43; 95% CI, 0.20–0.89) [38]. Furthermore, in the CREDENCE trial (Evaluation of the Effects of Canagliflozin on Renal and Cardiovascular Outcomes in Participants with Diabetic Nephropathy), canagliflozin was associated with a numerically lower risk of total stroke (HR, 0.77; 95% CI, 0.55–1.08), a result driven by a significantly lower risk of hemorrhagic stroke (HR, 0.50; 95% CI, 0.30–0.83) [39]. Finally, sotagliflozin reduced the risk of total stroke (HR, 0.66; 95% CI, 0.48–0.91) in the SCORED (Effect of Sotagliflozin on Cardiovascular and Renal Events in Patients with Type 2 Diabetes and Moderate Renal Impairment Who Are at Cardiovascular Risk) trial [40].

## 5. Stroke Risk Reduction by SGLT2 Inhibitors: Hints from Existing CVOTs

As mentioned previously, the meta-analyses of the CVOTs conducted in diabetic patients revealed that SGLT2 inhibitors as a class may have a neutral effect on fatal, non-fatal, and total stroke compared to placebos [20,21,22,41]. However, SGLT2 inhibitors might be associated with approximately 50% reductions with respect to the risk of hemorrhagic strokes [20,39]. Furthermore, SGLT2 inhibitors did reduce stroke risks in some groups of diabetic patients with CKD [22,42,43,44]. The effect of stroke risk reduction by sotagliflozin was demonstrated in diabetic patients with an estimated GFR of between 25 and 60 mL/min/1.73 m^2^ [40], and a similar effect by canagliflozin was observed in those with an estimated GFR < 45 mL/min/1.73 m^2^ [39].

### 5.1. The Role of Baseline Renal Function

As shown in Table 1, patients in the previous six CVOTs were generally comparable in their age, body mass index, baseline glycated hemoglobin, systolic blood pressure, and low-density lipoprotein cholesterol levels, as well as in the utilization of evidence-based cardiovascular medications such as renin–angiotensin–aldosterone system inhibitors and statins. On the other hand, the baseline eGFR was the lowest in the SCORED trial, followed by the CREDENCE trial. Coincidentally, the greatest reduction in total stroke risk was achieved by sotagliflozin in the SCORED trial (34%), followed by canagliflozin in the CREDENCE trial (23%). We speculated that the stroke prevention effects of SGLT2 inhibitors may differ for different renal function levels in diabetic patients, being more pronounced in those with more advanced renal impairment, e.g., eGFR < 45 mL/min/1.73 m^2^, via their effects on renal protection. Because standard-of-care medications such as statins may be less effective in preventing stroke in patients with advanced CKD [43], the benefits of SGLT2 inhibitors, despite the short period of follow-up in these CVOTs, stood out most prominently.

### 5.2. The Role of Atrial Fibrillation

Atrial fibrillation (AF) is a strong risk factor for strokes. Patients with either CKD or diabetes are more likely to have the comorbidity of AF [29]. In a multicenter observational study in Japan [45], both eGFR 45–59 and eGFR < 45 mL/min/1.73 m^2^ were associated with a 21% and 55% increase in the risk of cardioembolic ischemic stroke, respectively, and this result is probably related to the increased AF burden in the CKD population [46]. The meta-analyses of previous CVOTs have shown that SGLT2 inhibitors may be associated with a reduced risk of incident AF [21,39,47]. However, it remains unknown whether SGLT2 inhibitors reduced AF risks via their protective effects against heart failure and whether the risk of cardioembolic stroke was reduced as well among the users of SGLT2 inhibitors [47]. Notably, AF was not a pre-specified study endpoint in those CVOTs; therefore, the interpretation of these results should be performed cautiously. Further well-designed randomized trials are needed to answer these questions.

### 5.3. Differential Effect on Stroke Reduction by SGLT1/2 Dual Inhibition

Besides SGLT2, it is known that the SGLT1 in the brush border membrane of proximal renal tubules and small intestine is also responsible for glucose reabsorption [48,49,50]. Although SGLT2 mediates about 97% of glucose reabsorption in the human body, treatment with SGLT2 selective inhibitors only leads to a fractional glucose excretion of around 60%, probably related to the up-regulation of renal SGLT1 [50]. Table 2 lists the selectivity of the SGLT1/2 inhibition of currently available SGLT2 inhibitors. Among them, sotagliflozin has the highest selectivity for SGLT1 over SGLT2 (~0.05-fold), followed by canagliflozin (~0.004-fold). It has been hypothesized that the dual inhibition of both SGLT1 and SGLT2 by sotagliflozin may result in better glycemic control and lower total stroke risks than other SGLT2 inhibitors [48,49], although this result is obtained at the expense of an increased risk of diarrhea and volume depletion in the SCORED trial [40]. Moreover, SGLT1 was found to exist in the brain and the overexpression of SGLT1 in the area of brain damage may contribute to brain ischemia and reperfusion injury [51]. The inhibition of SGLT1 may thus have a neuroprotective effect by reducing the blood–brain barrier’s permeability and cerebral edema, thereby mitigating the stroke burden in diabetic patients with CKD [49,51].

### 5.4. Benefits of Reducing Hemorrhagic Stroke

Meta-analyses have shown that SGLT2 inhibitors may reduce the risk of hemorrhagic stroke risks by about 50% [20,39]. Several mechanisms may contribute to this beneficial effect. First, SGLT2 inhibitors lower blood pressure via diuretic effects related to glucosuria and natriuresis [52,53]. Since hypertension is a strong risk factor for hemorrhagic stroke [54], it is speculated that lowering blood pressure with SGLT2 inhibitors results in greater risk reductions in hemorrhagic stroke compared to other types of strokes. Second, SGLT2 inhibitors may reduce the risk of hemorrhagic stroke by increasing the levels of low-density lipoprotein cholesterol (LDL-C) and triglycerides. A large prospective cohort study enrolling 27,937 women found that LDL-C < 70 mg/dL and a low level of triglyceride were associated with hemorrhagic stroke [55]. Third, SGLT2 inhibitors might mitigate small vessel disease burden in the brain, such as cerebral microbleed (CMB), thus reducing the risk of hemorrhagic stroke.

Hemorrhagic strokes, such as ICH, used to be the most feared complication of antithrombotic therapy, including antiplatelet agents and oral anticoagulants. Among patients receiving antithrombotic therapy, those with CKD were more likely to have covert cerebral small vessel disease, including CMB [10]. The presence of CMB has been known to increase the risk of ICH after long-term antithrombotic therapy [56,57,58]. It is no wonder that all non-vitamin K oral anticoagulants include the baseline renal function as their dose reduction criteria [59]. Nevertheless, physicians still need to face the clinical dilemma of thrombosis risks versus bleeding risk when prescribing oral antiplatelet agents or anticoagulants for patients with CKD.

Given the significantly reduced risk of hemorrhagic stroke observed in the CVOTs of SGLT2 inhibitors, it may be anticipated that SGLT2 inhibitors will reduce the risk of ICH associated with long-term antithrombotic therapy in patients with CKD. However, further randomized clinical trials are needed to test this hypothesis.

## 6. Conclusions

We summarized the effect of SGLT2 inhibitors on stroke risk in diabetic patients with CKD and their potential roles in stopping the vicious cycle of cerebro–renal interactions. First, SGLT2 inhibitors have the effect of preventing the progression of CKD. This effect is independent of the diabetes status and baseline renal function [14,60]. Although not proven in randomized trials, it is reasonable to expect that stroke risks will decrease as the progression of CKD slows down. Second, at least in diabetic patients with CKD, SGLT2 inhibitors with partial SGLT1 inhibition, e.g., sotagliflozin and canagliflozin, may reduce the risk of total stroke, particularly in those with eGFR < 45 mL/min/1.73 m^2^ [39,40]. These findings are highly encouraging because traditional cardiovascular medications such as statins may have little benefits in the population with advanced CKD [43]. Finally, SGLT2 inhibitors may reduce the long-term risk of ischemic stroke via the reduction in incident AF, and it may reduce the risk of ICH complications when oral anticoagulants are used in CKD patients with AF via their potential effects on reducing the risk of hemorrhagic stroke.

Will SGLT2 inhibitors fulfill the long-standing unmet clinical need in diabetic patients with CKD and stroke? The cerebro–renal protective effects of SGLT2 inhibitors in stroke patients with CKD should be further tested in clinical trials. However, a simple search of the ongoing trials of SGLT2 inhibitors failed to find a trial of this kind [61]. Most published and ongoing trials of SGLT2 inhibitors focus on cardiorenal syndrome rather than cerebro–renal syndrome. Given the detrimental effect of stroke in patients with CKD, it should be reasonable to design and conduct such trials to generate evidence on this topic.

## Figures and Tables

**Table 1 ijms-24-00351-t001:** Baseline demographics and stroke outcomes in cardiovascular outcome trials of each sodium–glucose cotransporter-2 (SGLT2) inhibitor.

CVOT	EMPA-REG	CANVAS	DECLAIRE	CREDENCE	VERTIS-CV	SCORED
SGLT2 inhibitor	empagliflozin	canagliflozin	dapagliflozin	canagliflozin	ertugliflozin	sotagliflozin
Age	63 ± 9	63 ± 8	64 ± 7	63 ± 9	64 ± 8	69 (63–74) *
Male, %	71	64	63	66	70	55
Prior CVD, %	99	72	41	50	76	49
Prior stroke, %	23	19	8	21	21	9
BMI, mean	31 ± 5	32 ± 6	32 ± 6	31 ± 6	32 ± 5	32 (28–36) *
SBP, mmHg	135 ± 17	137 ± 16	135 ± 15	140 ± 16	134 ± 14	138 (127–149) *
HbA1c, %	8.1 ± 0.9	8.2 ± 0.9	8.3 ± 1.2	8.3 ± 1.3	8.2 ± 1.0	8.3 (7.6–9.3) *
LDL, mg/dL	86 ± 36	89 ± 35	89 ± 35	96 ± 41	89 ± 39	NR
eGFR	74 ± 22	77 ± 21	85 ± 16	56 ± 18	76 ± 21	44 (37–51) *
eGFR < 60, %	26	25	7	60	22	100
RAASi, %	81	80	81	100	81	89
Statin, %	81	75	75	70	82	NR
Total stroke	1.18 (0.89–1.56)	0.87 (0.69–1.09)	1.01 (0.84–1.21)	0.77 (0.55–1.08)	1.06 (0.82–1.37)	0.66 (0.52–0.89)
Non-fatal stroke	1.24 (0.92–1.67)	0.90 (0.71–1.15)	NR	0.97 (0.76–1.24)	1.00 (0.76–1.32)	NR
Fatal stroke	0.72 (0.33–1.55)	0.97 (0.52–1.83)	NR	0.77 (0.50–1.17)	NR	NR
Hemorrhagic stroke	0.64 (0.24–1.72)	0.50 (0.24–1.04)	NR	0.50 (0.30–0.83)	NR	NR

* Presented with median (interquartile range). BMI: body mass index; CVD: cardiovascular disease; CVOT: cardiovascular outcome trial; eGFR: estimated glomerular filtration rate; HbA1c: glycated hemoglobin; LDL: low-density lipoprotein; NR: not reported; RAASi: renin–angiotensin–aldosterone system inhibitor; SBP: systolic blood pressure; SGLT2, sodium–glucose cotransporter 2.

**Table 2 ijms-24-00351-t002:** Selectivity of sodium–glucose cotransporter 1/2 (SGLT1/2) inhibition.

	Empagliflozn	Canagliflozin	Dapaglifolozin	Ertugliflozin	Sotagliflozin
SGLT1 IC50	~8300	~710	~1400	~1960	~36
SGLT2 IC50	~3.1	~2.7	~1.2	~0.9	~1.8
Selectivity (SGLT1 over SGLT2)	~0.0004-fold	~0.004-fold	~0.0009-fold	~0.0005-fold	~0.05-fold

IC50: half maximal inhibitory concentration.

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
