# Peer review of "From Kidney Protection to Stroke Prevention: The Potential Role of Sodium Glucose Cotransporter-2 Inhibitors"

_ijms, 2022, doi:10.3390/ijms24010351_

Round 1

Reviewer 1 Report

The authors conducted a very comprehensive review of SGLT2 in cerebrovascular outcome. And the perspective is inspiring, especially the section of "The role of baseline renal function." Currently many research teams found the benefit of SGLT2 to reduce the hemorrhagic stroke. After reading this perspective, I think the readers will want to know more about the possible or speculative mechanism of this benefit, such as the animal studies result. The article will be better to briefly summarize some of the studies discussing this kind of mechanism.

And in table 1, the column of “eGFR< 60”, should add “, %” to be “eGFR< 60, %"

Author Response

We would like to express our sincere gratitude for your thorough consideration of our manuscript. Through the accurate and keen comments made by the reviewer, we were able to understand the critical points in our manuscript. For clarity, we quote the comments verbatim then add our responses below. 

Reviewer 1

The authors conducted a very comprehensive review of SGLT2 in cerebrovascular outcome. And the perspective is inspiring, especially the section of "The role of baseline renal function."

1. Currently many research teams found the benefit of SGLT2 to reduce the hemorrhagic stroke. After reading this perspective, I think the readers will want to know more about the possible or speculative mechanism of this benefit, such as the animal studies result. The article will be better to briefly summarize some of the studies discussing this kind of mechanism.

Response: Thank you for reminding us of the issue regarding the underlying mechanisms. Several mechanisms may contribute to this beneficial effect. First, SGLT2 inhibitors lower blood pressure via diuretic effects related to glucosuria and natriuresis [53,54]. Since hypertension is a strong risk factor for hemorrhagic stroke [55], it is speculated that blood pressure lowering with SGLT2 inhibitors results in greater risk reduction in hemorrhagic stroke, as compared to other types of stroke. Second, SGLT2 inhibitors may reduce the risk of hemorrhagic stroke by increasing levels of low-density lipoprotein cholesterol (LDL-C) and triglycerides. A large prospective cohort study enrolling 27937 women found that LDL-C < 70 mg/dL and a low level of triglyceride were associated with hemorrhagic stroke [56]. Third, SGLT2 inhibitors might mitigate small vessel disease burden in the brain, such as cerebral microbleed (CMB), thus reducing the risk of hemorrhagic stroke. Nevertheless, to the best of our knowledge, animal and human studies testing this hypothesis are lacking. We have discussed these mechanisms in our revised manuscript.

2. And in table 1, the column of “eGFR< 60”, should add “, %” to be “eGFR< 60, %"

Response: We revised this part of Table 1 accordingly.

Thank you again for all the valuable comments and help.

Reviewer 2 Report

The authors sent an article on a very novel and highly interesting topic. However, according to the title, they should have started with a small paragraph on kidney protection, which is the best known  (point four) and hence, to evolve towards the prevention of stroke, which would be the most innovative of the article and whose paragraph  (point three and 7), although the longest, lacks information. I advise you to reorganize the review.

Author Response

We would like to express our sincere gratitude for your thorough consideration of our manuscript. Through the accurate and keen comments made by the reviewer, we were able to understand the critical points in our manuscript. For clarity, we quote the comments verbatim then add our responses below.

Reviewer 2

The authors sent an article on a very novel and highly interesting topic. However, according to the title, they should have started with a small paragraph on kidney protection, which is the best known (point four) and hence, to evolve towards the prevention of stroke, which would be the most innovative of the article and whose paragraph (point three and 7), although the longest, lacks information. I advise you to reorganize the review.

Response: Thank you for this thoughtful comment. We have added a small paragraph on SGLT2 inhibition and kidney protection. We have also reorganized the manuscript following your suggestions.

Thank you again for all the valuable comments and help.
